NOS3 and CTH gene mutations as new molecular markers for detection of lung adenocarcinoma

Abdullah Ramadhan Iman 1
Rahman Sulaiman Luqman 2
Salihi Abbas abbas.salihi@su.edu.krd 1 3
1 Department of Biology, College of Science, Salahaddin University-Erbil , Erbil , Kurdistan Region , Iraq
2 Department of Medicine, College of Medicine, Hawler Medical University , Erbil , Kurdistan Region , Iraq
3 Center of Research and Strategic Studies, Lebanese French University , Erbil , Kurdistan Region , Iraq
Uversky Vladimir
Electronic publication date: 2023 Dec 12
Publication date: 2023
Volume: 11
Electronic Location ID: e16209
Received 2023 Jun 12; Accepted 2023 Sep 8
Copyright: ©2023 Abdullah Ramadhan et al.
Copyright year: 2023
Copyright holder: Abdullah Ramadhan et al.
License: This is an open access article distributed under the terms of the Creative Commons Attribution License, which permits unrestricted use, distribution, reproduction and adaptation in any medium and for any purpose provided that it is properly attributed. For attribution, the original author(s), title, publication source (PeerJ) and either DOI or URL of the article must be cited.
License URL: https://creativecommons.org/licenses/by/4.0/

Keywords: Lung adenocarcinoma, Nitric oxide, Hydrogen sulfide, Mutation, Mutation database

Funding: The authors received no funding for this work.

==============================
Gene mutations can contribute to lung adenocarcinoma (LUAD) development, metastasis, and therapy. This study aims to identify mutations in the endothelial nitric oxide synthase (eNOS or NOS3) and cystathionine γ-lyase (CSE or CTH) genes that are connected to LUAD symptoms. Two gene polymorphisms were identified using Sanger sequencing in 31 LUAD patients’ formalin-fixed paraffin-embedded (FFPE) tissues. Epidermal growth factor receptor (EGFR) mutation and programmed death-ligand 1 (PD-L1) expression were examined in 110 LUAD patients using real-time polymerase chain reaction and immunohistochemistry. Mutations in the selected genes were retrieved from the gnomAD database for all cancer types and the Mutagene and COSMIC databases for LUAD patients. The GeneMANIA prediction server was used to predict the interaction between the studied genes. Poorly and moderately differentiated tumours predominated, with pT3 N2 Mx being the most prevalent stage. Polymorphism data showed 189 NOS3 gene mutations and 34 CTH gene mutations. In 110 LUAD patients, 14 (12.73%) were PD-L1 positive and expressed 50% or more protein. Eight (7.27%) samples included EGFR mutations, including two deletions and two point mutations in exon 19, four point mutations in exon 21. In gnomAD, 4012 NOS3 mutations and 1214 CTH mutations are present. In the Mutagene and COSMIC databases, the NOS3 gene had 295 and 93 mutations, whereas the CTH gene had 61 and 36. According to the GeneMANIA prediction server, 10 genes are related to NOS3, eight with CTH, 15 with EGFR, and 5 with PD-L1. This study is the first to identify several previously unknown mutations in LUAD patients’ NOS3 and CTH genes, with potential therapeutic implications.

Introduction

Around 2.1 million new cases and 1.8 million deaths from lung cancer (LC) were reported in 2018; it is the leading cause of death worldwide (Sung et al., 2021) and it was the most common type of cancer in men diagnosed between 2013 and 2019 in both the Erbil and Duhok governorates of Iraq’s Kurdistan Region (M-Amen et al., 2022). The majority of these cases involved non-small cell lung cancer (NSCLC), which is divided into various histological categories. Lung adenocarcinoma (LUAD), the most prevalent histological subtype of LC, accounts for about 40% of all LC. Small airway epithelial, type II alveolar cells, which secrete mucus and other substances, give rise to LUAD (Anusewicz, Orzechowska & Bednarek, 2020; Denisenko, Budkevich & Zhivotovsky, 2018). The prognosis of LUAD remains elusive, with its 5-year survival rate being less than 20%, despite recent advancements in cancer treatments, including surgical resection, immunotherapy, chemotherapy, and radiotherapy (Song et al., 2021). Therefore, it is crucial to find more reliable biomarkers for assessing LUAD patients’ prognoses. Endothelial nitric oxide synthase, which is expressed in endothelial cells (Fish et al., 2005), produces endogenous gaseous molecules that have been recognized as gasotransmitters, including nitric oxide (NO), which is crucial for vasculogenesis (Förstermann & Sessa, 2012), tumor growth and invasion (Hu et al., 2020; Korde Choudhari et al., 2013), vasodilatation, neurotransmission and macrophage-mediated immunity (Xu et al., 2002). Human eNOS is encoded by the NOS3 gene, which spans 21 kb of genomic DNA and has 26 exons. It is located on chromosome 7q35-36 (Nassereddine et al., 2018). The most clinically significant polymorphisms in the NOS3 gene to date have been the 894G > T (rs1799983, G298D) (Karimi et al., 2022), intron VNTR 4a/b (a-deletion allele with 27 bp VNTR in intron 4) (Ramírez-Patiño et al., 2013; Sivri et al., 2014), and −786T > C (rs2070744) (Agúndez et al., 2020; Heidari, Khatami & Tahamtan, 2017). The NOS3 activity in LUAD samples was higher compared to other types of lung cancers and normal lung samples (Hiyoshi et al., 1992). Additionally, NOS activity was strongly linked to p53 gene mutation in stage I LUAD (Fujimoto et al., 1998). Specifically, the NOS3 894 G/T variant was significantly associated with EGFR mutation types of LUAD (Huang et al., 2018).

Like NO and carbon monoxide, hydrogen sulfide (H2S) is a gaseous signalling molecule and a member of the gasotransmitter family (Wang, 2003). Cystathionine β-synthase (CBS) and CSE are two pyridoxal 5′ phosphate-dependent enzymes that convert L-cysteine into endogenous H2S in mammalian tissues (Salihi et al., 2022). Endogenous H2S production is required for many bodily functions, including the respiratory system (Wang, 2012) several cancers, including colon cancer (Szabo et al., 2013), breast (Li et al., 2021), liver (Sun et al., 2021) and lung adenocarcinoma (Li et al., 2020) have also been linked to its development. CTH is mainly expressed in the liver, kidney, and vascular smooth muscles (Pan et al., 2012), it is one of the three enzymes in the transsulfuration pathway that is responsible for producing endogenous H2S (Kabil et al., 2011). The expression of CTH is regulated by several factors, and it is an inducible gene in numerous cell and tissue types. The human CTH gene’s core promoter region contains two consensus specificity protein 1 (Sp1) binding sites, and Sp1 activation in response to a number of stimuli induces CTH mRNA expression by binding to the human CTH promoter (Yang et al., 2011). Elevated expression of H2S-generating enzymes and increased H2S production were observed in several cultured LUAD cell lines (Yang et al., 2022). Inhibition of H2S-producing enzymes or reduction of intracellular H2S levels can induce mitochondrial dysfunction and enhance the sensitivity of LUAD cells to chemotherapeutic drugs (Szczesny et al., 2016). While there is not currently available study on the role and association of CTH gene polymorphisms in LUAD carcinogenesis. This study aims to show the contribution and effects of various gene polymorphisms in the NOS3 and CTH genes, as well as their associations with lung adenocarcinoma.

Materials & Methods

Human LUAD specimens

Thirty-one FFPE tissue samples of LUAD patients, of Kurdish nationality, were removed by microtome from patients who underwent surgical treatment. The microscopic morphology of tumor cells was used to classify lung cancer. The tumor node metastasis (TNM) staging system, 8th edition, was used in the staging method. Hematoxylin and eosin (H&E) staining, a regularly used method, was applied to analyze the structure and features of tissues. The T stage (size and extent of the main tumor), N stage (involvement of regional lymph nodes), and the existence of metastases (M stage) in lung cancer may all be confirmed by H&E staining (Lababede & Meziane, 2018). For this purpose, the ethical and regulatory issues related to human specimen collection for research purposes have been performed in compliance with the Declaration of Helsinki; approved by the Human Ethics Research Committee in the reference number (4/5/337) of the College of Science, Salahaddin University-Erbil. All patients were provided signed informed consent declaring to investigate the tissue materials. Patients with other co-morbid health problems which could introduce heterogeneity to the sample, such as additional acquired heart diseases, hypertension, diabetes mellitus, arthritis, ankylosing spondylitis, connective tissue diseases and other inflammatory diseases were excluded. Before the molecular investigation, tissue specimens were histologically confirmed. FFPE tissue samples were collected and stored at room temperature (20–22 °C) until subjected to DNA extraction and genotyping.

NOS3 and CTH mutation analysis by Sanger sequencing

The procedure of DNA extraction, amplification and sequencing has been previously described by Marangoni et al. (2008). In summary, using a commercial kit and following the manufacturer’s instructions, DNA was extracted and purified from 31 sectioned FFPE of LUAD tissue samples. Each sample block was 10 µm in diameter and divided into ten equal sections (ReliaPrepTM FFPE gDNA Miniprep System; Promega, Madison, WI, USA). The DNA content was then measured using a Thermo Scientific Nanodrop 1000 spectrophotometer, and the purity of the DNA was determined using an optimal A260/A280 absorbance ratio of 1.8. The study investigated two exons from the NOS3 gene (2 and 10) and two exons from the CTH gene (8 and 11). Purified DNA was amplified individually for each genetic variant using PCR on a Techne TC-5000 gradient thermal cycler (Cole-Parmer, Ltd.) using the primers listed below: NOS3 (first primer, exon 2) forward, AGG CCC TAT GGT AGT GCC TTT and reverse, TCT CTT AGT GCT GTG GTC AC; NOS3 (second primer, exon 10) AAG GCA GGA GAC AGT GGA TG and reverse, TGA AGG AAG AGT TCT GGT GGA; CTH (first primer, exon 8) forward, GGA CTT CTT GAG GAG TTG AAG C and reverse, ATT CTC ACC TCC TTC AGA GGC; and CTH (second primer, exon 11) forward, CTCCACTGACACAATATAGTGATCA and reverse, TGAAGGCAGCAGTAAGTCTTATT.

A ready-to-use master mix (Promega) has been used, containing Taq DNA polymerase, dNTPs, KCl, and reaction buffer. According to Abid, Qadir & Salihi (2021), the PCR thermocycling conditions consisted of the following: initial denaturation at 95 °C for 5 min, followed by 35 cycles of 95 °C for 30 s, annealing at 55 °C for 30 s, and elongation at 72 °C for 1 min, followed by a final extension step at 72 °C for 5 min. PCR products were separated using 2% agarose gel electrophoresis and compared with a 100 bp DNA marker (Froggabio, USA). The DNA marker included fragments of 400 bp from exon 8 of the CTH gene, 500 bp from exon 11 of the CTH gene, 300 bp from exon 2 of the NOS3 gene, and 600 bp from exon 10 of the NOS3 gene. Prior to casting into the tray, the gel was stained with safe dye (Safe DNA Gel Stain dye; Add Bio, Inc.). A gel documentation system (UV Transilluminator UST-20M-8K; Biostep GmbH) was used to visualise the gels.

Following the PCR procedure, the product has been sent to Macrogene, South Korea (using the same forward and reverse primer for each specified region in the gene) on an automated 3130 genetic analyzer (Applied Biosystems; Thermo Fisher Scientific, Inc.). Sequence data have been deposited to a curated data repository (GenBank database; http://www.ncbi.nlm.nih.gov/genbank/). The nucleotide sequence data stated are available in the GenBank database under the accession numbers OR750458.1 and OR757575. The Sanger sequencing data were analysed using the Mutation Surveyor software 5.1 (SoftGenetics, LLC) and compared to gnomAD, dbSNP, ClinVar, and COSMIC. The Mutation Surveyor programme results will be supplied as a file tree frame and a mutation report (output report table, ORT). A variety of ORTs can be generated using the “Reports” menu. An ORT indicates all of the detected mutations and connects them to their relevant sequence traces. One of the most crucial reports is the clinical report. It is a personalized report with choices for providing exclusively sample files with mutation calls or all locations with provided annotation information, including negative SNPs. For each identified mutation, the report shows one pair of sample files in a contig per page and provides the mutation code, Variant Percentage, and shared mutations in patients.

EGFR mutation and PD-L1 expression

For the EGFR mutation and PD-L1 expression tests, 110 FFPE samples of Kurdish nationality LUAD patients were used. Of these patients, 77 (70%) were men and 33 (30%) were women, respectively, with the following ages: Thirteen (11.82%) of the participants were under the age of fifty, followed by 29 (26.36%) in the 51–60 age range, 34 (30.91%) in the 61–70 age range, and 34 (30.91%) over the age of seventy.

According to the instructions in the product insert and as previously detailed by Angulo et al. (2012), the EGFR mutation was examined using the therascreen EGFR PCR kit (Qiagen, Inc., Tokyo, Japan) on the Rotor-Gene Q real-time instrument (Corbett Research, Qiagen GmbH). For each specimen, eight master mixes were prepared for testing on the Rotor-Gene Q machine. The real-time kit examines for deletions in exons 19 and 21 (it identifies the existence of deletions but is unable to differentiate between them). In addition to the mutation reactions, the kit provides a control gene reaction to assess the quality and quantity of the products. Real-time PCR testing takes around 4 hrs to complete. The EGFR gene testing methods have already been confirmed in the laboratory, and all reactions included negative and positive tissue controls.

As previously reported by Roach et al. (2016), automated staining techniques validated for the programmed death-ligand 1 (PD-L1) or cluster of differentiation 274 (CD274) immunohistochemistry (IHC) 22C3 pharmDx test were employed for IHC staining. In a 3-in-1 procedure, the PT Link (Dako PT100) was employed for deparaffinization, rehydration, and target retrieval. The specimens were first incubated with a monoclonal mouse anti-human PD-L1 antibody, clone 22C3, or a negative control reagent, mouse immunoglobulin G isotype control, before being exposed to an anti-mouse linker antibody specific to the primary antibody’s host species and a ready-to-use visualisation reagent composed of the secondary antibody and horseradish peroxidase molecules coupled to a dextran polymer back. Following the addition of 3,30-diaminobenzidine tetrahydrochloride chromogen, followed by 3,30-diaminobenzidine tetrahydrochloride enhancer, a visible reaction product precipitated at the antigen site. The specimens were then counterstained with hematoxylin and coverslipped. The data were examined using a light microscope. The sensitivity of the final test was optimised with the least amount of nonspecific staining by adjusting the primary antibody concentration and reagent incubation periods. Positivity was defined as entire circumferential or partial cell membrane staining of live tumour cells with 1+ to 3+ intensity to determine PD-L1 protein expression. Nonspecific staining was rated in increments of 0.25 on a 0 to 3 intensity scale. PD-L1 was not used to identify immune cells associated with tumours. Cytoplasmic staining was disregarded when computing the score if it was present. The score (TPS) was calculated using the percentage of PD-L1-positive tumour cells compared to total tumour cells in the denominator. The negative control reagent must result in no specific membrane staining and nonspecific (nonmembrane) staining of 1+ intensity in LUAD specimens.

Mutation retrieval from databases and interaction between genes

Mutations in the selected genes across various cancer types were retrieved using the Genome Aggregation Database (gnomAD) versions 2.1.1 and 3.1.2. (https://gnomad.broadinstitute.org/). The gnomAD database is an important tool for genetic variation and cancer research. It contains global exome and genome sequencing data, and multiple large-scale sequencing programs give enormous variability in genomes among populations (Gudmundsson et al., 2022; Koch, 2020).

The entire reported mutations of selected genes have been retrieved from the Mutagene (developed at Panchenko Research Group, NCBI) (https://www.ncbi.nlm.nih.gov/research/mutagene/gene) and “catalogue of somatic mutations in cancer” (COSMIC v96, released May 31, 2022) (https://cancer.sanger.ac.uk/cosmic) in order to analyze the mutations commonly found in LUAD patients. As previously stated by (Goncearenco et al., 2017), Mutagene comprises 9,450 cancer samples from 37 projects containing 1,139,534 non-recurring mutations. Single base substitutions in protein-coding genomic loci from whole genome and whole exome-sequenced samples from ICGC projects, TCGA, and the Paediatric Cancer Genome Project are considered more frequently sequenced genes or mutations identified by genotyping. Cancer Drivers, Potential Drivers, and Passengers are predicted by Mutagene based on Bscore thresholds defined using a comprehensive benchmark dataset comprised of multiple experimental studies. MutaGene adjusts the frequency of recurrent mutations in cancer patients according to their mutability concerning the predicted background mutagenesis. As a consequence, the computations will be affected by both the mutational model at baseline and the cohort. Catalogue of Somatic Mutations in Cancer (COSMIC; freely accessible to researchers worldwide) is a comprehensive database that covers cancer somatic mutations. It includes clinical and pathological data on mutation site, frequency, and function. This database may discover common mutations and other genetic variations that may contribute to cancer formation or progression and serve as therapeutic targets (Tate et al., 2019).

NOS3, CTH, EGFR, and PD-L1 interactions were predicted using the GeneMANIA prediction tool (https://genemania.org/). This online database provides gene interactions through the use of protein-protein interaction networks, co-expression patterns, and genetic connections. These four genes’ projected connections may reveal their involvement in cellular processes and disease pathways and lead to novel therapeutic targets or diagnostic biomarkers. The GeneMANIA prediction server is a major step in understanding gene interactions and their effects on human disease and health (Franz et al., 2018; Warde-Farley et al., 2010).

Results

Patient characteristics

Fifteen of the LUAD patients are female (48.39%) and 16 are male (51.61%), with the following ages: Three (9.68%) were under 50, seven (22.58%) were between 51 and 60, eighteen (58.06%) were between 61 and 70, and three (9.68%) were older than 71.

Patients with LUAD were shown to have a very differentiated distribution, with several different subgroups. A greater number of cases were categorized as either poorly differentiated or moderately poorly differentiated (38.01%). These results suggest that aggressive and poorly differentiated tumors are significantly overrepresented among the patients. The frequency of moderately differentiated adenocarcinomas was lower than that of poorly differentiated subtypes, accounting for 14.28% of all cases. In addition, the study population had a lower incidence of primary adenocarcinoma and papillary adenocarcinoma (4.76%).

The extent of tumor invasion, lymph node involvement in the regional area, and the presence of distant metastases were all revealed by TNM staging. For 44.44% of cases, the TNM staging was determined to be pT3 N2 Mx. According to the results of this study, lymph nodes are affected by a significant proportion of locally advanced cancers. pT2 N1 Mx was the second most common staging pattern, occurring in 22.22% of patients and indicating tumors that were localised to the primary site but had spread to nearby lymph nodes. In addition, 11.11% of patients classified to each of the substages pT2 N2 Mx, pT2 Nx Mx, and T2 NO Mx. The differences between the phases are indicative of factors such tumor size, lymph node involvement, and the presence or absence of distant metastases, as shown in Table 1.

Table 1 Clinicopathologic characteristics of LUAD patients.

Characteristic	Percentage	
Sex		
Female	48.39%	
Male	51.61%	
Age (year)		
>50	9.68%	
51–60	22.58%	
61–70	58.06%	
<71	58.06%	
Degree of differentiation		
Poorly differentiated	38.01%	
Moderately-poorly differentiated	38.01%	
Moderately differentiated	14.28%	
Primary adenocarcinoma	4.76%	
Papillary adenocarcinoma	4.76%	
TNM staging		
pT3 N2 Mx.	44.44%	
pT2 N1 Mx	22.22%	
pT2 N2 Mx	11.11%	
pT2 Nx Mx.	11.11%	
T2 NO Mx	11.11%	

Mutation analysis

At various locations on chromosome 7, a total of 189 mutations in the NOS3 gene were identified in LUAD patients. Among them, there were 174 substitution mutations, 155 heterozygous and 19 homozygous mutations, and eight deletions, five insertions and two duplications were not recorded, these mutations were changed 24 different types of amino acids. Furthermore, except for four variant mutations, the mutations were never recorded in databases. Six homozygous substitutions in the NOS3 gene account for the greatest percentage of variants (100%); amino acid changes have occurred place in five of them, but only one of them has been reported in external databases.

On chromosome 1, the CTH gene was examined and a total of 31 different mutations were identified in various locations. There were 27 heterozygous mutations, three homozygous mutations, and one deletion among (28658delA). Furthermore, except for one variant mutation (28400G < GT), the mutations have never been documented in the databases. Furthermore, 6 of the newly found mutant variants in the CTH gene are associated with a change in amino acid level. Twenty-two heterozygous and one homozygous substitutions were identified in the CTH gene, but no amino acid changes were detected in any of them, nor were any of them reported in external databases, as shown in Fig. 1 and Table S1, and as summarised in Table 2 and Fig. 2.

Figure 1 DNA sequence electropherograms demonstrating nucleotide mutations in the NOS3 and CTH genes.

(A–D) NOS3 homozygous T–G substitution, heterozygous deletion, heterozygous duplication, and heterozygous insertion. (E and F) Heterozygous substitution from A to AG and deletion 2865C of the CTH gene.

Table 2 A summary of the variants identified in the NOS3 and CTH genes by mutation DNA variant analysis in LUAD patients.

Genes	Number of mutations	Types of mutations	Unchanged amino acid	Amino acid changes	
NOS3	189	Substitution	136 heterozygous (2 in ED)	19 heterozygous (0 in ED)	
			5 homozygous (1 in ED)	14 homozygous (1 in ED)	
		Deletion	8 (0 in ED)		
		Insertion	5 (0 in ED)		
		Duplication	2 (0 in ED)		
CTH	31	Substitution	22 heterozygous (0 in ED)	5 heterozygous (1 in ED)	
			1 homozygous (0 in ED)	2 homozygous (0 in ED)	
		Deletion	1 (0 in ED)		
Notes.

ED external databases

Figure 2 A flowchart summarizing the variations recognized within the NOS3 and CTH genes by mutation DNA variant analysis in LUAD patients.

EGFR mutation and PD-L1 expression

There were 14 (12.73%) samples with PD-L1 positive that expressed 50% or more PD-L1 protein among the 110 LUAD patients. While eight (7.27%) of the samples had EGFR mutations, including two tumors with exon 19 deletions, two tumors with exon 19 point mutations, four tumors with exon 21 point mutation (c.2573T > G p. Leu858Arg), and two tumors with exon 21 point mutation (p.L858R), as shown in Table S5.

Database mutations and genes interactions

The total reported mutations of all cancer types acquired by gnomAD revealed that the NOS3 gene has 4012 mutations, while the CTH gene has 1214 mutations, as shown in Table S2.

The retrieved gene mutations in the Mutagene (including mutations in the selected cancer cohort, all genome wide-studies in ICGC, and all samples in COSMIC) and COSMIC databases in LUAD patients were further analyzed for the selected genes, as shown in Tables S3 and S4. The NOS3 gene had the greatest number of mutations (295 and 93 in both databases, respectively), while the CTH gene had 61 and 36 mutations. The most common type of mutations found in the NOS3 gene in LUAD patients in the Mutagene database is missense followed by silent mutations, and their status is mostly passenger mutations, followed by potential driver and driver mutations, whereas the mutations in the CTH gene are mostly missense followed by silent and nonsense mutations, and their status is mostly mutations, passenger mutations, and one potential driver. Substitution-missense mutations are the most common type of mutation found in LUAD patients in the NOS3 gene in the COSMIC database, they are followed by substitution-coding silent, unknown, and substitution-nonsense mutations, while substitution-missense, unknown, substitution-coding silent and substitution-nonsense mutations are the most common types of mutations in the CTH gene. The data retrieved from the gnomAD, Mutagene and COSMIC databases are summarized in Fig. 3.

Figure 3 (A–C) A flowchart summarizing the dataretrieved from the gnomAD, Mutagene and COSMIC databases.

According to the GeneMANIA prediction server, there are 10 genes (AKT1, CAV1, CDO1, EPS8, HSP90AB1, LYPLA1, NOSIP, NOSTRIN, WASL and ZDHHC21) associated with the NOS3 gene, eight genes (CAV1, CBS, CDO1, DARS1, EGFR, MTHFD1, PTPN1 and SQOR) associated with the CTH gene, 15 genes (AKT1, ANXA2, CAV1, CBS, CDO1, CSK, DARS1, EGF, EPS8, HSP90AB1, NOSIP, PIK3CA, PTPN1, WASL and ZDHHC21) associated with the EGFR gene, and five genes (CAV1, CSK, NOSTRIN, PDCD1 and PTPN1) associated with the PD-L1 gene via co-expression, co-localization, genetic interaction, pathway, physical interaction, predicted, or shared protein domain (Fig. 4 and Table S6). Only the CAV1 gene is shared between the four gene pathways, and eight genes (AKT1, CAV1, CDO1, EPS8, HSP90AB1, NOSIP, WASL, and ZDHHC21) are shared between the NOS3 and EGFR genes, indicating that these two genes are closely related.

Figure 4 The interaction between NOS3, CTH, EGFR and PD-L1 genes and their related genes retrieved from the GeneMANIA prediction server.

Discussion

The current study sheds light on the TNM staging and differentiation levels of adenocarcinoma patients. The aggressiveness of adenocarcinomas is highlighted by their observed differentiation grade distribution, with a large percentage falling into the poorly differentiated and moderately poorly differentiated groups. Tumours with poor differentiation tend to be more aggressive and have a worse prognosis. The samples from LUAD patients were also categorised using the TNM system for staging cancer. These specimens of tissues were classified as T2 or T3, suggesting a tumour size between 3 and 7 cm. In addition, the majority of patients were categorised as N1 or N2, indicating that the cancer cells were located in the lymph nodes inside the lungs or in the area where the lungs link to the airway (the hilum), whereas they were categorised as MX, indicating that the degree of metastatic spread was difficult to determine.

The impact of NOS3 gene polymorphism on cancer is currently a topic of debate, mainly due to factors such as sample size, cancer type, and the influence of ethnic variation (Nan et al., 2019). Gao et al. (2015) conducted a meta-analysis study and found significant associations between the −786T > C polymorphism in the NOS3 gene and breast cancer risk, as well as between the 894G > T polymorphism and risk of all other types of cancer. In contrast, another meta-analysis conducted by Haque et al. (2015) found no significant association between the single-nucleotide polymorphism(SNP) 894 G > T in the NOS3 gene and overall cancer susceptibility. The current study identified that the NOS3 gene had the highest number of missense substitution mutations. This finding aligns with data from the Mutagene and COSMIC databases, which also indicated that the majority of substitutional mutations in patients with LUAD were missense, while the remaining mutations were either nonsense or silent. Bayraktutan et al. (2020) investigated the association between ADMA levels, NO, and polymorphisms in LC patients. They also examined the role of the G894T and T-786C polymorphisms in the development of lung cancer. Fujita et al. (2010) investigated the association between the G894T polymorphism (rs1799983) located in exon 7 of a gene, resulting in an amino acid change from G298D, and its potential impact on NSCLC patient survival. They found that this polymorphism is linked to diminished basal NO production. Huang et al. (2018) conducted a study examining the connection between NOS3 genetic polymorphisms (-786T/C and 894 G/T) and EGFR mutations in patients with LUAD.

The current study found that 30 out of the 34 mutations detected in the CTH gene are heterozygous mutations. This finding aligns with the data from the Mutagene and COSMIC databases, which indicate that the majority of substitutional mutations observed in LUAD patients are missense mutations, with the remaining being nonsense and silent mutations. This study is the first to investigate LUAD patients specifically, as prior research has primarily examined the impact of CTH polymorphism on the development and progress of cardiac diseases. Several studies have reported associations between polymorphisms in the CTH gene and variations in plasma total homocysteine concentrations. Wang & Hegele (2003); Wang et al. (2004); Zhu, Lin & Banerjee (2008) found that these polymorphisms are linked to either increased or decreased plasma total homocysteine concentrations. Additionally, certain polymorphisms in the CTH gene have been linked to cardiac diseases. In particular, Söderström et al. (2022), found that the minor alleles of CTH G1208T are associated with a higher risk of fatal myocardial infarction in women. Zhou et al. (2020) reported that the indel polymorphism rs113044851 is linked to susceptibility to sudden cardiac death. Furthermore, Mrozikiewicz et al. (2015) noticed a connection between the rs482843 polymorphism and the risk of preeclampsia. Giannakopoulou et al. (2019) reported no association between the CTH 1364 G > T polymorphism and coronary artery disease. Similarly, Li et al. (2008) found no association between the SNPs rs482843 and rs1021737 in the CTH gene and essential hypertension.

From the results of the 189 mutations observed in the NOS3 gene, 33 were considered critical as they resulted in a replacement in one of the amino acids produced. Among these critical replacement, glycine changes were the most prevalent, with nine occurrences. This was followed by mutations in glutamine (five replacement), leucine (four replacement), proline (four replacement), and aspartic acid (four replacement). Out of the 34 replacements observed in the CTH gene, only seven results in changes to the amino acid type. The most common alteration involves a substitution of serine amino acids. While there is currently no existing study investigating the consequences of this change, our study’s findings present an opportunity for researchers in the field to explore potential side effects. Specifically, it is possible that this change could impair the catalytic activity of the NOS3 and CTH enzymes. The mutation of a serine residue near the catalytic site of choline acetyltransferase in zebrafish embryos significantly impairs the enzyme’s motility (Joshi et al., 2018). Therefore, the finding of novel NOS3 and CTH gene mutations in LUAD patients could have clinical consequences. These mutations can serve as biomarkers for risk assessment, early diagnosis, prognosis, and therapeutic response (Koçer et al., 2020; Kraus et al., 2009). Studying the genetic landscape of LUAD could help in the development of targeted therapies and individualised treatments. However, it is essential to observe that the provided results lack information regarding the functional significance of the mutations. Nevertheless, by integrating patient characteristics and mutational analysis, we can understand the relationship between genetic variations and demographic characteristics of LUAD patients. As an example, the higher prevalence of LUAD among older people may be linked to the aggressiveness of adenocarcinomas and greater accumulation of genetic mutations over time. In addition, genetic variations, such as those identified in the NOS3 and CTH genes, might influence the development and progression of LUAD in both male and female patients. This is consistent with the mutations previously identified by our group in colorectal cancer patients (Housein, Kareem & Salihi, 2021), in which several NOS3 and CTH gene mutations were discovered. Again, in this study, we analyzed and identified mutations in the NOS3 and CTH genes in LUAD patients, providing a strong basis for future research into the disease’s underlying causes. In addition, these results imply the possibility of previously unreported mutations being confirmed at a particular location on both chromosomes.

LUAD with EGFR mutations account for 10–15% of cases, and these mutations are associated with heightened sensitivity to epidermal growth factor receptor (EGFR) tyrosine kinase inhibitors (TKIs), such as gefitinib, erlotinib, and osimertinib. EGFR TKIs have been demonstrated to increase overall survival and progression-free survival in LUAD patients with EGFR mutations. In order to guide treatment decisions and improve patient outcomes, EGFR mutation testing is essential (Wu et al., 2019). Exons 18, 19, 20, and/or 21 are the most potential sites for EGFR mutations. Exon-19 deletion (Del19) and exon-21 mutation are the two most common activating mutations (Liu et al., 2020). Moreover, PD-L1 expression is a biomarker for the effectiveness of immune checkpoint inhibitors such as pembrolizumab, nivolumab, and atezolizumab (Mathew, Safyan & Shu, 2017). When PD-L1 expression is high, patients with LUAD receiving these medications have higher response rates and better overall survival. Testing for PD-L1 expression is essential to guide therapeutic choices and improve patient outcomes (Garon et al., 2015). PD-L1 protein expression detected by IHC analysis for LUAD has been the primary predictive biomarker studied for response to anti-PD-1/PD-L1 immunotherapy (Yu et al., 2016). Only one study (Huang et al., 2018) found that eNOS 894 G/T variants were significantly associated with EGFR mutation types of lung adenocarcinoma, specifically exon 19 in-frame deletion and that this could be used to predict tumor invasiveness and response to therapy.

Conclusions

In conclusion, the genetic variations and clinical data of LUAD patients, including patient features, tumour differentiation, and TNM staging, depict an individual picture. A better understanding of this complex disease could result in more effective personalised therapies and improved patient outcomes. This is the first study to identify an important number of previously unreported mutations in the NOS3 and CTH genes in LUAD patients. By analysing mutations in the NOS3, CTH, and EGFR genes and integrating their findings with the most pertinent mutation-related datasets, this study identifies the molecular mechanisms underlying the development and progression of LUAD. Due to the small sample size and Sanger sequencing, the study’s results may be limited. This study is crucial to LUAD research and establishes the framework for future studies. This could be done by conducting functional studies to understand the molecular effects of these mutations, investigating their association with clinical outcomes in larger cohorts, exploring their predictive value for treatment response, expanding the genomic analysis to identify additional mutations and variations and integrating multi-omics data to understand their functional implications could be done.

Supplemental Information

Table S1 Supplementary Table 1

Variants identified in NOS3 and CTH genes LUAD patients analyzed with mutation DNA variant analysis.

Click here for additional data file.

Table S2 Supplementary Table 2

Summary of mutations in the NOS3 and CTH genes in all types of cancer retrieved from the gnomAD database.

Click here for additional data file.

Table S3 Supplementary Table 3

Summary of mutations in the NOS3 and CTH genes in lung adenocarcinoma retrieved from the Mutagene database.

Click here for additional data file.

Table S4 Supplementary Table 4

Summary of mutations in the NOS3 and CTH genes in lung adenocarcinoma retrieved from the COSMIC database.

Click here for additional data file.

Table S5 Supplementary table 5

EGFR mutation and PD-L1 expression in patient samples with LUAD.

Click here for additional data file.

Table S6 Supplementary table 6

The interaction between NOS3, CTH, EGFR and PD-L1 gene and their related genes retrieved from GeneMania database

Click here for additional data file.

Special thanks to all oncologists, the Histopathology department, and laboratory staff members at Nanakaly Hospital, Rizgary Hospital, Par Hospital, Tamara and Luay medical lab in Erbil, Iraq.

Additional Information and Declarations

Competing Interests

Author Contributions

Human Ethics

DNA Deposition

Data Availability

The authors declare there are no competing interests.

Iman Abdullah Ramadhan performed the experiments, analyzed the data, prepared figures and/or tables, authored or reviewed drafts of the article, and approved the final draft.

Luqman Rahman Sulaiman conceived and designed the experiments, authored or reviewed drafts of the article, and approved the final draft.

Abbas Salihi conceived and designed the experiments, analyzed the data, prepared figures and/or tables, authored or reviewed drafts of the article, and approved the final draft.

The following information was supplied relating to ethical approvals (i.e., approving body and any reference numbers):

The Human Research Ethics Committee of the College of Science, Salahaddin University-Erbil approved the study (4-5-337).

The following information was supplied regarding the deposition of DNA sequences:

The sequences are available at GenBank: OR750458.1 and OR757575.

The following information was supplied regarding data availability:

The data are available at Figshare: Salihi, Abbas (2022). NOS3 & CTH Lung cancer. figshare. Journal contribution. https://doi.org/10.6084/m9.figshare.21706031.v1.

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
