# Peer review of "NOS3 and CTH gene mutations as new molecular markers for detection of lung adenocarcinoma"

_PeerJ, doi:10.7717/peerj.16209_

## Round 0.1 · original submission · Major Revisions

Please address the concerns of both reviewers and amend the manuscript accordingly.

Reviewer 1 ·

Basic reporting

1. Language: The paper is written in a professional English language. However, there are still several ambiguities or typos. For example, “FFPE” is unclear in line 23. The sentence in line 236-238 is long and confusing. “clearer” in line 367 sounds confusing, which should be “clear” as I guess.
2. Abstract and Introduction: Abstract contains a lot of detailed information, which should be summarized so that key points of this paper can be highlighted. Introduction part gives good introduction of lung cancer, NOS3 and CTH gene. However, there is a lack of introducing the relation between NOS3/CTH genes and lung cancer. It’s unclear that why the authors come out this idea. It would be better if more background/thoughts can be provided.
3. Table: Table 2 is too long, which should be put in supplementary materials.

Experimental design

Methods are provided with sufficient details.

Validity of the findings

In line 335-336, the authors state that “the study's focus on the NOS3 and CTH genes may provide additional insight into the molecular mechanisms underlying LUAD.” Please further elaborate the “molecular mechanisms” that NOS3 and CTH genes adopt to contribute to LUAD. If not, this argument is over stated.

Reviewer 2 ·

Basic reporting

The manuscript submitted by I.A. Ramazhan et al. identifies the new mutations in the endothelial nitric oxide synthase (eNOS or NOS3) and cystathionine-lyase (CSE or CTH) genes that are linked to various Lung adenocarcinoma (LUAD) symptoms. Sanger sequencing was performed to identify gene polymorphisms and RT-PCR and immunohistochemistry were used to examine epidermal growth factor receptor (EGFR) mutation and programmed death-ligand 1 (PDL1) expression in LUAD patients. The databases used in the studies were gnomAD, Mutagene, COSMIC, and GeneMANIA for mutation and interaction studies. The three major conclusions drawn from studies are as follows:
1. The polymorphism data revealed 189 mutations in the NOS3 gene and 34 mutations in the CTH gene.
2. For gene NOS3, there are 4012, 295, and 93 mutations were observed using different databases using gnomAD, Mutagene, and COSMIC, respectively. For the CTH gene, 1214, 61, and 36 mutations were observed using different databases using gnomAD, Mutagene, and COSMIC, respectively.
3. According to the GeneMANIA prediction server, there were ten genes associated with the NOS3 gene, eight genes with the CTH gene, fifteen genes with the EGFR gene, and five genes associated with the PD-L1 gene.

Experimental design

• The abstract, material and methods, results, discussion, and conclusion are well-written with well-designed experiments. The references are cited well by the authors.
• Some of the experiments mentioned in the method and not shown in the results such as immunohistochemistry experiments on FFPF tissue samples of the LUAD patients. It will be helpful for the readers if the data is included (maybe in the supplementary data). These experiments also shed light on the protein level presence of the genes of interest in LUAD patients.
• It would be helpful for the readers if the authors could include the flow charts for the database results to show which database showed how many mutations in the genes NOS3 and CTH. In the same flowchart, it could also be mentioned how many mutations are already known and how many new mutations are observed.
• However, the manuscript has some major and minor issues (mentioned in the additional comments) that should be addressed before this work is publishable.

Validity of the findings

The identification of mutations would serve as a groundwork for future research in this area but it is difficult to understand the clinical implications of the new mutations (shown in the manuscript) in the NOS3 and CTH genes present in the LUAD patients. Mutations in the gene could result in the gain of unwanted function or loss of the native function of the proteins. If authors could wisely choose some of the mutations and show their effect, that research could be included in the manuscript. This would also differentiate their work from the previous research.

Additional comments

Specific points are given below:
1. It would be helpful to the reader if the authors included the flowchart for some of the results obtained using databases (gnomAD, Mutagene, and COSMIC) and mentioned the number of mutations already known in literature and how many are new mutations.
2. Line 53, I would recommend using the word “elusive” from “frustrating” in the sentence.
3. Line 80-81, please include references in the sentence “There have been very few studies published on the role and relationship of CTH gene polymorphisms in adenocarcinoma carcinogenesis.” Also, mention how your work fills in the gap or adds more information to the known data.
4. I would recommend the authors to work on the discussion section to connect the previous information with the results obtained. It reads like the known literature is given as a bullet point but not connected well with the obtained results. This will also make the conclusion well-ordered and connected.

---

## Round 0.2 · Minor Revisions

Please address the remaining concerns of the reviewer and amend the manuscript accordingly.

Reviewer 1 ·

Basic reporting

no comment

Experimental design

no comment

Validity of the findings

no comment

Additional comments

All the questions have been well addressed. With new figures added and language polished, the paper is substantially improved. I would recommend acceptance.

Reviewer 2 ·

Basic reporting

All my concerns have been addressed, and the manuscript has been revised accordingly. Now the manuscript reads well, and Figures 2 and 3 make it easy for the reader to understand the new findings. The discussion and conclusion are cohesively written, and I thank the authors for the wonderful job. I recommend publication of this revised manuscript in PeerJ with some very minor suggestions as follows:
1. In line 194, change “have been” to “were” to keep the grammar consistent throughout the manuscript.
2. In line 251, please correct/ complete the sentence.
3. In line 319, please change Glu298Asp to G298D for consistency, as the single letter code is used in the manuscript.
4. Include a reference for the sentence in lines 376-377 about the PD-L1 expression and checkpoint inhibitors.
5. In line 257, the sentence “Nevertheless, by integrating patient…characteristics of LUAD patients”, it is possible to change the word “figure out” to “understand”.
6. Proofreading of the manuscript is needed to correct some extra brackets here and there in the manuscript. (such as line 262)

Experimental design

Discussed in the 1st revision

Validity of the findings

Discussed in the 1st revision

Additional comments

N/A

---

## Round 0.3 · accepted · Accept

All remaining issues pointed out by the reviewer were adequately addressed and the amended manuscript is acceptable now.